# BRUNO: A Deep Recurrent Model for Exchangeable Data

**Iryna Korshunova** ♥
Ghent University
iryna.korshunova@ugent.be

**Jonas Degrave** ♥ †
Ghent University
jonas.degrave@ugent.be

**Ferenc Huszár**
Twitter
fhuszar@twitter.com

**Yarin Gal** ♠
University of Oxford
yarin@cs.ox.ac.uk

**Arthur Gretton** ♠
Gatsby Unit, UCL
arthur.gretton@gmail.com

**Joni Dambre** ♠
Ghent University
joni.dambre@ugent.be

## Abstract

We present a novel model architecture which leverages deep learning tools to perform exact Bayesian inference on sets of high dimensional, complex observations. Our model is provably exchangeable, meaning that the joint distribution over observations is invariant under permutation: this property lies at the heart of Bayesian inference. The model does not require variational approximations to train, and new samples can be generated conditional on previous samples, with cost linear in the size of the conditioning set. The advantages of our architecture are demonstrated on learning tasks that require generalisation from short observed sequences while modelling sequence variability, such as conditional image generation, few-shot learning, and anomaly detection.

## 1 Introduction

We address the problem of modelling unordered sets of objects that have some characteristic in common. Set modelling has been a recent focus in machine learning, both due to relevant application domains and to efficiency gains when dealing with groups of objects [5, 18, 20, 23]. The relevant concept in statistics is the notion of an exchangeable sequence of random variables – a sequence where any re-ordering of the elements is equally likely. To fulfil this definition, subsequent observations must behave like previous ones, which implies that we can make predictions about the future. This property allows the formulation of some machine learning problems in terms of modelling exchangeable data. For instance, one can think of few-shot concept learning as learning to complete short exchangeable sequences [10]. A related example comes from the generative image modelling field, where we might want to generate images that are in some ways similar to the ones from a given set. At present, however, there are few flexible and provably exchangeable deep generative models to solve this problem.

Formally, a finite or infinite sequence of random variables $x_1, x_2, x_3, \ldots$ is said to be exchangeable if for all $n$ and all permutations $\pi$

$$p(x_1, \ldots, x_n) = p\left(x_{\pi(1)}, \ldots, x_{\pi(n)}\right), \tag{1}$$

i. e. the joint probability remains the same under any permutation of the sequence. If random variables in the sequence are independent and identically distributed (i. i. d.), then it is easy to see that the sequence is exchangeable. The converse is false: exchangeable random variables can be correlated. One example of an exchangeable but non-i. i. d. sequence is a sequence of variables $x_1, \ldots, x_n$, which

---

♥♠Equal contribution †Now at DeepMind.

jointly have a multivariate normal distribution $\mathcal{N}_n(\mathbf{0}, \mathbf{\Sigma})$ with the same variance and covariance for all the dimensions [1]: $\Sigma_{ii} = 1$ and $\Sigma_{ij,i\neq j} = \rho$, with $0 \leq \rho < 1$.

The concept of exchangeability is intimately related to Bayesian statistics. De Finetti's theorem states that every exchangeable process (infinite sequence of random variables) is a mixture of i. i. d. processes:

$$p(x_1, \ldots, x_n) = \int p(\theta) \prod_{i=1}^{n} p(x_i|\theta)d\theta, \tag{2}$$

where $\theta$ is some parameter (finite or infinite dimensional) conditioned on which, the random variables are i. i. d. [1]. In our previous Gaussian example, one can prove that $x_1, \ldots, x_n$ are i. i. d. with $x_i \sim \mathcal{N}(\theta, 1 - \rho)$ conditioned on $\theta \sim \mathcal{N}(0, \rho)$.

In terms of predictive distributions $p(x_n|x_{1:n-1})$, the stochastic process in Eq. 2 can be written as

$$p(x_n|x_{1:n-1}) = \int p(x_n|\theta)p(\theta|x_{1:n-1})d\theta, \tag{3}$$

by conditioning both sides on $x_{1:n-1}$. Eq. 3 is exactly the posterior predictive distribution, where we marginalise the likelihood of $x_n$ given $\theta$ with respect to the posterior distribution of $\theta$. From this follows one possible interpretation of the de Finetti's theorem: learning to fit an exchangeable model to sequences of data is implicitly the same as learning to reason about the hidden variables behind the data.

One strategy for defining models of exchangeable sequences is through explicit Bayesian modelling: one defines a prior $p(\theta)$, a likelihood $p(x_i|\theta)$ and calculates the posterior in Eq. 2 directly. Here, the key difficulty is the intractability of the posterior and the predictive distribution $p(x_n|x_{1:n-1})$. Both of these expressions require integrating over the parameter $\theta$, so we might end up having to use approximations. This could violate the exchangeability property and make explicit Bayesian modelling difficult.

On the other hand, we do not have to explicitly represent the posterior to ensure exchangeability. One could define a predictive distribution $p(x_n|x_{1:n-1})$ directly, and as long as the process is exchangeable, it is consistent with Bayesian reasoning. The key difficulty here is defining an easy-to-calculate $p(x_n|x_{1:n-1})$ which satisfies exchangeability. For example, it is not clear how to train or modify an ordinary recurrent neural network (RNN) to model exchangeable data. In our opinion, the main challenge is to ensure that a hidden state contains information about all previous inputs $x_{1:n}$ regardless of sequence length.

In this paper, we propose a novel architecture which combines features of the approaches above, which we will refer to as BRUNO: Bayesian RecUrrent Neural mOdel. Our model is *provably exchangeable*, and makes use of deep features learned from observations so as to model complex data types such as images. To achieve this, we construct a *bijective* mapping between random variables $x_i \in \mathcal{X}$ in the observation space and features $z_i \in \mathcal{Z}$, and explicitly define an exchangeable model for the sequences $z_1, z_2, z_3, \ldots$, where we know an analytic form of $p(z_n|z_{1:n-1})$ without explicitly computing the integral in Eq. 3.

Using BRUNO, we are able to generate samples conditioned on the input sequence by sampling directly from $p(x_n|x_{1:n-1})$. The latter is also tractable to evaluate, i. e. has linear complexity in the number of data points. In respect of model training, evaluating the predictive distribution requires a single pass through the neural network that implements $\mathcal{X} \mapsto \mathcal{Z}$ mapping. The model can be learned straightforwardly, since $p(x_n|x_{1:n-1})$ is differentiable with respect to the model parameters.

The paper is structured as follows. In Section 2 we will look at two methods selected to highlight the relation of our work with previous approaches to modelling exchangeable data. Section 3 will describe BRUNO, along with necessary background information. In Section 4, we will use our model for conditional image generation, few-shot learning, set expansion and set anomaly detection. Our code is available at `github.com/IraKorshunova/bruno`.

## 2 Related work

Bayesian sets [6] aim to model exchangeable sequences of binary random variables by analytically computing the integrals in Eq. 2, 3. This is made possible by using a Bernoulli distribution for the

likelihood and a beta distribution for the prior. To apply this method to other types of data, e.g. images, one needs to engineer a set of binary features [7]. In that case, there is usually no one-to-one mapping between the input space $\mathcal{X}$ and the features space $\mathcal{Z}$: in consequence, it is not possible to draw samples from $p(x_n|x_{1:n-1})$. Unlike Bayesian sets, our approach does have a bijective transformation, which guarantees that inference in $\mathcal{Z}$ is equivalent to inference in space $\mathcal{X}$.

The neural statistician [5] is an extension of a variational autoencoder model [8, 15] applied to datasets. In addition to learning an approximate inference network over the latent variable $z_i$ for every $x_i$ in the set, approximate inference is also implemented over a latent variable $c$ – a context that is global to the dataset. The architecture for the inference network $q(c|x_1, \ldots, x_n)$ maps every $x_i$ into a feature vector and applies a mean pooling operation across these representations. The resulting vector is then used to produce parameters of a Gaussian distribution over $c$. Mean pooling makes $q(c|x_1, \ldots, x_n)$ invariant under permutations of the inputs. In addition to the inference networks, the neural statistician also has a generative component $p(x_1, \ldots, x_n|c)$ which assumes that $x_i$'s are independent given $c$. Here, it is easy to see that $c$ plays the role of $\theta$ from Eq. 2. In the neural statistician, it is intractable to compute $p(x_1, \ldots, x_n)$, so its variational lower bound is used instead. In our model, we perform an implicit inference over $\theta$ and can exactly compute predictive distributions and the marginal likelihood. Despite these differences, both neural statistician and BRUNO can be applied in similar settings, namely few-shot learning and conditional image generation, albeit with some restrictions, as we will see in Section 4.

## 3 Method

We begin this section with an overview of the mathematical tools needed to construct our model: first the Student-t process [17]; and then the Real NVP – a deep, stably invertible and learnable neural network architecture for density estimation [4]. We next propose BRUNO, wherein we combine an exchangeable Student-t process with the Real NVP, and derive recurrent equations for the predictive distribution such that our model can be trained as an RNN. Our model is illustrated in Figure 1.

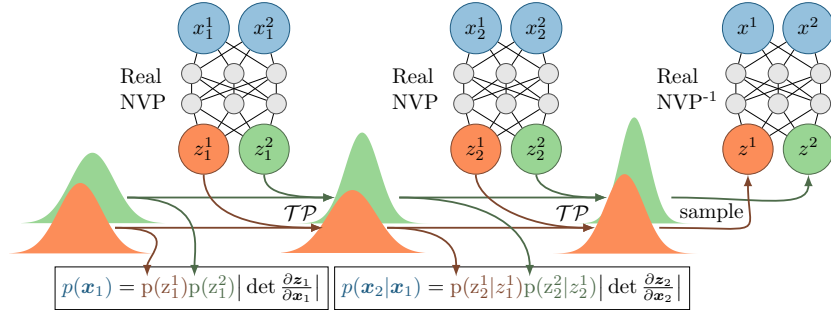

Figure 1: A schematic of the BRUNO model. It depicts how Bayesian thinking can lead to an RNN-like computational graph in which Real NVP is a bijective feature extractor and the recurrence is represented by Bayesian updates of an exchangeable Student-t process.

### 3.1 Student-t processes

The Student-t process ($\mathcal{TP}$) is the most general elliptically symmetric process with an analytically representable density [17]. The more commonly used Gaussian processes ($\mathcal{GP}$s) can be seen as limiting case of $\mathcal{TP}$s. In what follows, we provide the background and definition of $\mathcal{TP}$s.

Let us assume that $z = (z_1, \ldots z_n) \in \mathbb{R}^n$ follows a multivariate Student-t distribution $MVT_n(\nu, \mu, K)$ with degrees of freedom $\nu \in \mathbb{R}_+ \setminus [0, 2]$, mean $\mu \in \mathbb{R}^n$ and a positive definite $n \times n$ covariance matrix $K$. Its density is given by

$$p(z) = \frac{\Gamma(\frac{\nu+n}{2})}{((\nu-2)\pi)^{n/2}\Gamma(\nu/2)}|K|^{-1/2}\left(1 + \frac{(z-\mu)^T K^{-1}(z-\mu)}{\nu-2}\right)^{-\frac{\nu+n}{2}}. \tag{4}$$

For our problem, we are interested in computing a conditional distribution. Suppose we can partition $\boldsymbol{z}$ into two consecutive parts $\boldsymbol{z}_a \in \mathbb{R}^{n_a}$ and $\boldsymbol{z}_b \in \mathbb{R}^{n_b}$, such that

$$\begin{bmatrix} \boldsymbol{z}_a \\ \boldsymbol{z}_b \end{bmatrix} \sim MVT_n\left(\nu, \begin{bmatrix} \boldsymbol{\mu}_a \\ \boldsymbol{\mu}_b \end{bmatrix}, \begin{bmatrix} \boldsymbol{K}_{aa} & \boldsymbol{K}_{ab} \\ \boldsymbol{K}_{ba} & \boldsymbol{K}_{bb} \end{bmatrix}\right). \tag{5}$$

Then conditional distribution $p(\boldsymbol{z}_b|\boldsymbol{z}_a)$ is given by

$$\begin{aligned}
p(\boldsymbol{z}_b|\boldsymbol{z}_a) &= MVT_{n_b}\left(\nu + n_a, \tilde{\boldsymbol{\mu}}_{\boldsymbol{b}}, \frac{\nu + \beta_a - 2}{\nu + n_a - 2}\tilde{\boldsymbol{K}}_{bb}\right), \\
\tilde{\boldsymbol{\mu}}_{\boldsymbol{b}} &= \boldsymbol{K}_{ba}\boldsymbol{K}_{aa}^{-1}(\boldsymbol{z}_a - \boldsymbol{\mu}_a) + \boldsymbol{\mu}_b \\
\beta_a &= (\boldsymbol{z}_a - \boldsymbol{\mu}_a)^T \boldsymbol{K}_{aa}^{-1}(\boldsymbol{z}_a - \boldsymbol{\mu}_a) \\
\tilde{\boldsymbol{K}}_{bb} &= \boldsymbol{K}_{bb} - \boldsymbol{K}_{ba}\boldsymbol{K}_{aa}^{-1}\boldsymbol{K}_{ab}.
\end{aligned} \tag{6}$$

In the general case, when one needs to invert the covariance matrix, the complexity of computing $p(\boldsymbol{z}_b|\boldsymbol{z}_a)$ is $\mathcal{O}(n_a^3)$. These computations become infeasible for large datasets, which is a known bottleneck for $\mathcal{GP}$s and $\mathcal{TP}$s [13]. In Section 3.3, we will show that exchangeable processes do not have this issue.

The parameter $\nu$, representing the degrees of freedom, has a large impact on the behaviour of $\mathcal{TP}$s. It controls how heavy-tailed the t-distribution is: as $\nu$ increases, the tails get lighter and the t-distribution gets closer to the Gaussian. From Eq. 6, we can see that as $\nu$ or $n_a$ tends to infinity, the predictive distribution tends to the one from a $\mathcal{GP}$. Thus, for small $\nu$ and $n_a$, a $\mathcal{TP}$ would give less certain predictions than its corresponding $\mathcal{GP}$.

A second feature of the $\mathcal{TP}$ is the scaling of the predictive variance with a $\beta_a$ coefficient, which explicitly depends on the values of the conditioning observations. From Eq. 6, the value of $\beta_a$ is precisely the Hotelling statistic for the vector $\boldsymbol{z}_a$, and has a $\chi^2_{n_a}$ distribution with mean $n_a$ in the event that $\boldsymbol{z}_a \sim \mathcal{N}_{n_a}(\boldsymbol{\mu}_a, \boldsymbol{K}_{aa})$. Looking at the weight $(\nu+\beta_a-2)/(\nu+n_a-2)$, we see that the variance of $p(\boldsymbol{z}_b|\boldsymbol{z}_a)$ is increased over the Gaussian default when $\beta_a > n_a$, and is reduced otherwise. In other words, when the samples are dispersed more than they would be under the Gaussian distribution, the predictive uncertainty is increased compared with the Gaussian case. It is helpful in understanding these two properties to recall that the multivariate Student-t distribution can be thought of as a Gaussian distribution with an inverse Wishart prior on the covariance [17].

## 3.2    Real NVP

Real NVP [4] is a member of the normalising flows family of models, where some density in the input space $\mathcal{X}$ is transformed into a desired probability distribution in space $\mathcal{Z}$ through a sequence of invertible mappings [14]. Specifically, Real NVP proposes a design for a bijective function $f : \mathcal{X} \mapsto \mathcal{Z}$ with $\mathcal{X} = \mathbb{R}^D$ and $\mathcal{Z} = \mathbb{R}^D$ such that **(a)** the inverse is easy to evaluate, i.e. the cost of computing $\boldsymbol{x} = f^{-1}(\boldsymbol{z})$ is the same as for the forward mapping, and **(b)** computing the Jacobian determinant takes linear time in the number of dimensions $D$. Additionally, Real NVP assumes a simple distribution for $\boldsymbol{z}$, e.g. an isotropic Gaussian, so one can use a change of variables formula to evaluate $p(\boldsymbol{x})$:

$$p(\boldsymbol{x}) = p(\boldsymbol{z})\left|\det\left(\frac{\partial f(\boldsymbol{x})}{\partial \boldsymbol{x}}\right)\right|. \tag{7}$$

The main building block of Real NVP is a coupling layer. It implements a mapping $\mathcal{X} \mapsto \mathcal{Y}$ that transforms half of its inputs while copying the other half directly to the output:

$$\begin{cases} \boldsymbol{y}^{1:d} = \boldsymbol{x}^{1:d} \\ \boldsymbol{y}^{d+1:D} = \boldsymbol{x}^{d+1:D} \odot \exp(s(\boldsymbol{x}^{1:d})) + t(\boldsymbol{x}^{1:d}), \end{cases} \tag{8}$$

where $\odot$ is an elementwise product, $s$ (scale) and $t$ (translation) are arbitrarily complex functions, e.g. convolutional neural networks.

One can show that the coupling layer is a bijective, easily invertible mapping with a triangular Jacobian and composition of such layers preserves these properties. To obtain a highly nonlinear mapping $f(\boldsymbol{x})$, one needs to stack coupling layers $\mathcal{X} \mapsto \mathcal{Y}_1 \mapsto \mathcal{Y}_2 \cdots \mapsto \mathcal{Z}$ while alternating the dimensions that are being copied to the output.

To make good use of modelling densities, the Real NVP has to treat its inputs as instances of a continuous random variable [19]. To do so, integer pixel values in $\boldsymbol{x}$ are dequantised by adding uniform noise $\boldsymbol{u} \in [0,1)^D$. The values $\boldsymbol{x} + \boldsymbol{u} \in [0,256)^D$ are then rescaled to a $[0,1)$ interval and transformed with an elementwise function: $f(x) = \text{logit}(\alpha + (1-2\alpha)x)$ with some small $\alpha$.

### 3.3   BRUNO: the exchangeable sequence model

We now combine Bayesian and deep learning tools from the previous sections and present our model for exchangeable sequences whose schematic is given in Figure 1.

Assume we are given an exchangeable sequence $\boldsymbol{x}_1, \ldots, \boldsymbol{x}_n$, where every element is a D-dimensional vector: $\boldsymbol{x}_i = (x_i^1, \ldots x_i^D)$. We apply a Real NVP transformation to every $\boldsymbol{x}_i$, which results in an exchangeable sequence in the latent space: $\boldsymbol{z}_1, \ldots, \boldsymbol{z}_n$, where $\boldsymbol{z}_i \in \mathbb{R}^D$. The proof that the latter sequence is exchangeable is given in Appendix A.

We make the following assumptions about the latents:

**A1**: dimensions $\{z^d\}_{d=1,\ldots,D}$ are independent, so $p(\boldsymbol{z}) = \prod_{d=1}^D p(z^d)$

**A2**: for every dimension $d$, we assume the following: $(z_1^d, \ldots z_n^d) \sim MVT_n(\nu^d, \mu^d \mathbf{1}, \boldsymbol{K}^d)$, with parameters:

- degrees of freedom $\nu^d \in \mathbb{R}_+ \setminus [0,2]$
- mean $\mu^d \mathbf{1}$ is a $1 \times n$ dimensional vector of ones multiplied by the scalar $\mu^d \in \mathbb{R}$
- $n \times n$ covariance matrix $\boldsymbol{K}^d$ with $\boldsymbol{K}_{ii}^d = v^d$ and $\boldsymbol{K}_{ij,i\neq j}^d = \rho^d$ where $0 \leq \rho^d < v^d$ to make sure that $\boldsymbol{K}^d$ is a positive-definite matrix that complies with covariance properties of exchangeable sequences [1].

The exchangeable structure of the covariance matrix and having the same mean for every $n$, guarantees that the sequence $z_1^d, z_2^d \ldots z_n^d$ is exchangeable. Because the covariance matrix is simple, we can derive recurrent updates for the parameters of $p(z_{n+1}^d | z_{1:n}^d)$. Using the recurrence is a lot more efficient compared to the closed-form expressions in Eq. 6 since we want to compute the predictive distribution for every step $n$.

We start from a prior Student-t distribution for $p(z_1)$ with parameters $\mu_1 = \mu$, $v_1 = v$, $\nu_1 = \nu$, $\beta_1 = 0$. Here, we will drop the dimension index $d$ to simplify the notation. A detailed derivation of the following results is given in Appendix B. To compute the degrees of freedom, mean and variance of $p(z_{n+1}|z_{1:n})$ for every $n$, we begin with the recurrent relations

$$\nu_{n+1} = \nu_n + 1, \quad \mu_{n+1} = (1-d_n)\mu_n + d_n z_n, \quad v_{n+1} = (1-d_n)v_n + d_n(v-\rho), \quad (9)$$

where $d_n = \frac{\rho}{v+\rho(n-1)}$. Note that the $\mathcal{GP}$ recursions simply use the latter two equations, i.e. if we were to assume that $(z_1^d, \ldots z_n^d) \sim \mathcal{N}_n(\mu^d \mathbf{1}, \boldsymbol{K}^d)$. For $\mathcal{TP}$s, however, we also need to compute $\beta$ – a data-dependent term that scales the covariance matrix as in Eq. 6. To update $\beta$, we introduce recurrent expressions for the auxiliary variables:

$$\tilde{z}_i = z_i - \mu$$

$$a_n = \frac{v+\rho(n-2)}{(v-\rho)(v+\rho(n-1))}, \quad b_n = \frac{-\rho}{(v-\rho)(v+\rho(n-1))}$$

$$\beta_{n+1} = \beta_n + (a_n - b_n)\tilde{z}_n^2 + b_n \left(\sum_{i=1}^n \tilde{z}_i\right)^2 - b_{n-1}\left(\sum_{i=1}^{n-1} \tilde{z}_i\right)^2.$$

From these equations, we see that computational complexity of making predictions in exchangeable $\mathcal{GP}$s or $\mathcal{TP}$s scales linearly with the number of observations, i.e. $\mathcal{O}(n)$ instead of a general $\mathcal{O}(n^3)$ case where one needs to compute an inverse covariance matrix.

So far, we have constructed an exchangeable Student-t process in the latent space $\mathcal{Z}$. By coupling it with a bijective Real NVP mapping, we get an exchangeable process in space $\mathcal{X}$. Although we do not have an explicit analytic form of the transitions in $\mathcal{X}$, we still can sample from this process and evaluate the predictive distribution via the change of variables formula in Eq. 7.

### 3.4 Training

Having an easy-to-evaluate autoregressive distribution $p(\boldsymbol{x}_{n+1}|\boldsymbol{x}_{1:n})$ allows us to use a training scheme that is common for RNNs, i.e. maximise the likelihood of the next element in the sequence at every step. Thus, our objective function for a single sequence of fixed length $N$ can be written as $\mathcal{L} = \sum_{n=0}^{N-1} \log p(\boldsymbol{x}_{n+1}|\boldsymbol{x}_{1:n})$, which is equivalent to maximising the joint log-likelihood $\log p(\boldsymbol{x}_1, \ldots, \boldsymbol{x}_N)$. While we do have a closed-form expression for the latter, we chose not to use it during training in order to minimize the difference between the implementation of training and testing phases. Note that at test time, dealing with the joint log-likelihood would be inconvenient or even impossible due to high memory costs when $N$ gets large, which again motivates the use of a recurrent formulation.

During training, we update the weights of the Real NVP model and also learn the parameters of the prior Student-t distribution. For the latter, we have three trainable parameters per dimension: degrees of freedom $\nu^d$, variance $v^d$ and covariance $\rho^d$. The mean $\mu^d$ is fixed to 0 for every $d$ and is not updated during training.

## 4 Experiments

In this section, we will consider a few problems that fit naturally into the framework of modeling exchangeable data. We chose to work with sequences of images, so the results are easy to analyse; yet BRUNO does not make any image-specific assumptions, and our conclusions can generalise to other types of data. Specifically, for non-image data, one can use a general-purpose Real NVP coupling layer as proposed by Papamakarios et al. [12]. In contrast to the original Real NVP model, which uses convolutional architecture for scaling and translation functions in Eq. 8, a general implementation has $s$ and $t$ composed from fully connected layers. We experimented with both convolutional and non-convolutional architectures, the details of which are given in Appendix C.

In our experiments, the models are trained on image sequences of length 20. We form each sequence by uniformly sampling a class and then selecting 20 random images from that class. This scheme implies that a model is trained to implicitly infer a class label that is global to a sequence. In what follows, we will see how this property can be used in a few tasks.

### 4.1 Conditional image generation

We first consider a problem of generating samples conditionally on a set of images, which reduces to sampling from a predictive distribution. This is different from a general Bayesian approach, where one needs to infer the posterior over some meaningful latent variable and then 'decode' it.

To draw samples from $p(\boldsymbol{x}_{n+1}|\boldsymbol{x}_{1:n})$, we first sample $\boldsymbol{z} \sim p(\boldsymbol{z}_{n+1}|\boldsymbol{z}_{1:n})$ and then compute the inverse Real NVP mapping: $\boldsymbol{x} = f^{-1}(\boldsymbol{z})$. Since we assumed that dimensions of $\boldsymbol{z}$ are independent, we can sample each $z^d$ from a univariate Student-t distribution. To do so, we modified Bailey's polar t-distribution generation method [2] to be computationally efficient for GPU. Its algorithm is given in Appendix D.

In Figure 2, we show samples from the prior distribution $p(\boldsymbol{x}_1)$ and conditional samples from a predictive distribution $p(\boldsymbol{x}_{n+1}|\boldsymbol{x}_{1:n})$ at steps $n = 1, \ldots, 20$. Here, we used a convolutional Real NVP model as a part of BRUNO. The model was trained on Omniglot [10] same-class image sequences of length 20 and we used the train-test split and preprocessing as defined by Vinyals et al. [21]. Namely, we resized the images to $28 \times 28$ pixels and augmented the dataset with rotations by multiples of 90 degrees yielding 4,800 and 1,692 classes for training and testing respectively.

To better understand how BRUNO behaves, we test it on special types of input sequences that were not seen during training. In Appendix E, we give an example where the same image is used throughout the sequence. In that case, the variability of the samples reduces as the models gets more of the same input. This property does not hold for the neural statistician model [5], discussed in Section 2. As mentioned earlier, the neural statistician computes the approximate posterior $q(\boldsymbol{c}|\boldsymbol{x}_1, \ldots, \boldsymbol{x}_n)$ and then uses its mean to sample $\boldsymbol{x}$ from a conditional model $p(\boldsymbol{x}|\boldsymbol{c}_{mean})$. This scheme does not account for the variability in the inputs as a consequence of applying mean pooling over the features of $\boldsymbol{x}_1, \ldots, \boldsymbol{x}_n$ when computing $q(\boldsymbol{c}|\boldsymbol{x}_1, \ldots, \boldsymbol{x}_n)$. Thus, when all $x_i$'s are the same, it would still sample different instances from the class specified by $x_i$. Given the code provided by the authors of

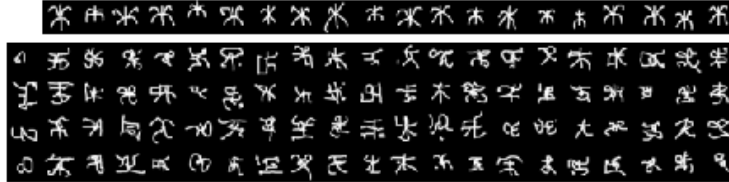

Figure 2: Samples generated conditionally on the sequence of the unseen Omniglot character class. An input sequence is shown in the top row and samples in the bottom 4 rows. Every column of the bottom subplot contains 4 samples from the predictive distribution conditioned on the input images up to and including that column. That is, the 1st column shows samples from the prior $p(\boldsymbol{x})$ when no input image is given; the 2nd column shows samples from $p(\boldsymbol{x}|\boldsymbol{x}_1)$ where $\boldsymbol{x}_1$ is the 1st input image in the top row and so on.

the neural statistician and following an email exchange, we could not reproduce the results from their paper, so we refrained from making any direct comparisons.

More generated samples from convolutional and non-convolutional architectures trained on MNIST [11], Fashion-MNIST [22] and CIFAR-10 [9] are given in the appendix. For a couple of these models, we analyse the parameters of the learnt latent distributions (see Appendix F).

## 4.2 Few-shot learning

Previously, we saw that BRUNO can generate images of the unseen classes even after being conditioned on a couple of examples. In this section, we will see how one can use its conditional probabilities not only for generation, but also for a few-shot classification.

We evaluate the few-shot learning accuracy of the model from Section 4.1 on the unseen Omniglot characters from the 1,692 testing classes following the $n$-shot and $k$-way classification setup proposed by Vinyals et al. [21]. For every test case, we randomly draw a test image $\boldsymbol{x}_{n+1}$ and a sequence of $n$ images from the target class. At the same time, we draw $n$ images for every of the $k-1$ random decoy classes. To classify an image $\boldsymbol{x}_{n+1}$, we compute $p(\boldsymbol{x}_{n+1}|\boldsymbol{x}_{1:n}^{C=i})$ for each class $i = 1 \ldots k$ in the batch. An image is classified correctly when the conditional probability is highest for the target class compared to the decoy classes. This evaluation is performed 20 times for each of the test classes and the average classification accuracy is reported in Table 1.

For comparison, we considered three models from Vinyals et al. [21]: **(a)** k-nearest neighbours (k-NN), where matching is done on raw pixels (Pixels), **(b)** k-NN with matching on discriminative features from a state-of-the-art classifier (Baseline Classifier), and **(c)** Matching networks.

We observe that BRUNO model from Section 4.1 outperforms the baseline classifier, despite having been trained on relatively long sequences with a generative objective, i.e. maximising the likelihood of the input images. Yet, it cannot compete with matching networks – a model tailored for a few-shot learning and trained in a discriminative way on short sequences such that its test-time protocol exactly matches the training time protocol. One can argue, however, that a comparison between models trained generatively and discriminatively is not fair. Generative modelling is a more general, harder problem to solve than discrimination, so a generatively trained model may waste a lot of statistical power on modelling aspects of the data which are irrelevant for the classification task. To verify our intuition, we fine-tuned BRUNO with a discriminative objective, i.e. maximising the likelihood of correct labels in $n$-shot, $k$-way classification episodes formed from the training examples of Omniglot. While we could sample a different $n$ and $k$ for every training episode like in matching networks, we found it sufficient to fix $n$ and $k$ during training. Namely, we chose the setting with $n = 1$ and $k = 20$. From Table 1, we see that this additional discriminative training makes BRUNO competitive with state-of-the-art models across all $n$-shot and $k$-way tasks.

As an extension to the few-shot learning task, we showed that BRUNO could also be used for online set anomaly detection. These experiments can be found in Appendix H.

Table 1: Classification accuracy for a few-shot learning task on the Omniglot dataset.

| Model | 5-way | | 20-way | |
|---|---|---|---|---|
| | 1-shot | 5-shot | 1-shot | 5-shot |
| PIXELS [21] | 41.7% | 63.2% | 26.7% | 42.6% |
| BASELINE CLASSIFIER [21] | 80.0% | 95.0% | 69.5% | 89.1% |
| MATCHING NETS [21] | 98.1% | 98.9% | 93.8% | 98.5% |
| BRUNO | 86.3% | 95.6% | 69.2% | 87.7% |
| BRUNO (discriminative fine-tuning) | 97.1% | 99.4% | 91.3% | 97.8% |

### 4.3 $\mathcal{GP}$-based models

In practice, we noticed that training $\mathcal{TP}$-based models can be easier compared to $\mathcal{GP}$-based models as they are more robust to anomalous training inputs and are less sensitive to the choise of hyperparameters. Under certain conditions, we were not able to obtain convergent training with $\mathcal{GP}$-based models which was not the case when using $\mathcal{TP}$s; an example is given in Appendix G. However, we found a few heuristics that make for a successful training such that $\mathcal{TP}$ and $\mathcal{GP}$-based models perform equally well in terms of test likelihoods, sample quality and few-shot classification results. For instance, it was crucial to use weight normalisation with a data-dependent initialisation of parameters of the Real NVP [16]. As a result, one can opt for using $\mathcal{GP}$s due to their simpler implementation. Nevertheless, a Student-t process remains a strictly richer model class for the latent space with negligible additional computational costs.

## 5 Discussion and conclusion

In this paper, we introduced BRUNO, a new technique combining deep learning and Student-t or Gaussian processes for modelling exchangeable data. With this architecture, we may carry out implicit Bayesian inference, avoiding the need to compute posteriors and eliminating the high computational cost or approximation errors often associated with explicit Bayesian inference.

Based on our experiments, BRUNO shows promise for applications such as conditional image generation, few-shot concept learning, few-shot classification and online anomaly detection. The probabilistic construction makes the BRUNO approach particularly useful and versatile in transfer learning and multi-task situations. To demonstrate this, we showed that BRUNO trained in a generative way achieves good performance in a downstream few-shot classification task without any task-specific retraining. Though, the performance can be significantly improved with discriminative fine-tuning.

Training BRUNO is a form of meta-learning or learning-to-learn: it learns to perform Bayesian inference on various sets of data. Just as encoding translational invariance in convolutional neural networks seems to be the key to success in vision applications, we believe that the notion of exchangeability is equally central to data-efficient meta-learning. In this sense, architectures like BRUNO and Deep Sets [23] can be seen as the most natural starting point for these applications.

As a consequence of exchangeability-by-design, BRUNO is endowed with a hidden state which integrates information about all inputs regardless of sequence length. This desired property for meta-learning is usually difficult to ensure in general RNNs as they do not automatically generalise to longer sequences than they were trained on and are sensitive to the ordering of inputs. Based on this observation, the most promising applications for BRUNO may fall in the many-shot meta-learning regime, where larger sets of data are available in each episode. Such problems naturally arise in privacy-preserving on-device machine learning, or federated meta-learning [3], which is a potential future application area for BRUNO.

## Acknowledgements

We would like to thank Lucas Theis for his conceptual contributions to BRUNO, Conrado Miranda and Frederic Godin for their helpful comments on the paper, Wittawat Jitkrittum for useful discussions, and Lionel Pigou for setting up the hardware.

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
