[Supplementary Material]

## A    Proofs

### Lemma 1

*Given an exchangeable sequence $(x_1, x_2, \ldots, x_n)$ of random variables $x_i \in \mathcal{X}$ and a bijective mapping $f : \mathcal{X} \mapsto \mathcal{Z}$, the sequence $(f(x_1), f(x_2), \ldots, f(x_n))$ is exchangeable.*

**Proof.** Consider a vector function $\boldsymbol{g} : \mathbb{R}^n \mapsto \mathbb{R}^n$ such that $(x_1, \ldots, x_n) \mapsto (z_1 = f(x_1), \ldots, z_n = f(x_n))$. A change of variable formula gives:

$$p(x_1, x_2, \ldots, x_n) = p(z_1, z_2, \ldots, z_n) \left| \det \boldsymbol{J} \right|,$$

where $\det \boldsymbol{J} = \prod_{i=1}^{n} \frac{\partial f(x_i)}{\partial x_i}$ is the determinant of the Jacobian of $\boldsymbol{g}$. Since both the joint probability of $(x_1, x_2, \ldots, x_n)$ and the $\left| \det \boldsymbol{J} \right|$ are invariant to the permutation of sequence entries, so must be $p(z_1, z_2, \ldots, z_n)$. This proves that $(z_1, z_2, \ldots, z_n)$ is exchangeable. $\square$

### Lemma 2

*Given two exchangeable sequence $\boldsymbol{x} = (x_1, x_2, \ldots, x_n)$ and $\boldsymbol{y} = (y_1, y_2, \ldots, y_n)$ of random variables, where $x_i$ is independent from $y_j$ for $\forall i, j$, the concatenated sequence $\boldsymbol{x}^\frown \boldsymbol{y} = ((x_1, y_1), (x_2, y_2), \ldots, (x_n, y_n))$ is exchangeable as well.*

**Proof.** For any permutation $\pi$, as both sequences $\boldsymbol{x}$ and $\boldsymbol{y}$ are exchangeable we have:

$$p(x_1, x_2, \ldots, x_n)p(y_1, y_2, \ldots, y_n) = p(x_{\pi(1)}, x_{\pi(2)}, \ldots, x_{\pi(n)})p(y_{\pi(1)}, y_{\pi(2)}, \ldots, y_{\pi(n)}).$$

Independence between elements in $\boldsymbol{x}$ and $\boldsymbol{y}$ allows to write it as a joint distribution:

$$p((x_1, y_1), (x_2, y_2) \ldots, (x_n, y_n)) = p((x_{\pi(1)}, y_{\pi(1)}), (x_{\pi(2)}, y_{\pi(2)}), \ldots, (x_{\pi(n)}, y_{\pi(n)})),$$

and thus the sequence $\boldsymbol{x}^\frown \boldsymbol{y}$ is exchangeable. $\square$

This Lemma justifies our construction with $D$ independent exchangeable processes in the latent space as given in A1 from Section 3.3.

## B    Derivation of recurrent Bayesian updates for exchangeable Student-t and Gaussian processes

We assume that $\boldsymbol{x} = (x_1, x_2, \ldots x_n) \in \mathbb{R}^n$ follows a multivariate Student-t distribution $MVT_n(\nu, \boldsymbol{\mu}, \boldsymbol{K})$ with degrees of freedom $\nu \in \mathbb{R}_+ \setminus [0, 2]$, mean $\boldsymbol{\mu} \in \mathbb{R}^n$ and a positive definite $n \times n$ covariance matrix $\boldsymbol{K}$. Its density is given by:

$$p(\boldsymbol{x}) = \frac{\Gamma(\frac{\nu+n}{2})}{((\nu-2)\pi)^{n/2}\Gamma(\nu/2)}|\boldsymbol{K}|^{-1/2}\left(1 + \frac{(\boldsymbol{x}-\boldsymbol{\mu})^T\boldsymbol{K}^{-1}(\boldsymbol{x}-\boldsymbol{\mu})}{\nu-2}\right)^{-\frac{\nu+n}{2}}. \tag{1}$$

Note that this parameterization of the multivariate t-distribution as defined by Shah et al. [8] is slightly different from the commonly used one. We used this parametrization as it makes the formulas simpler.

If we partition $\boldsymbol{x}$ into two consecutive parts $\boldsymbol{x}_a \in \mathbb{R}^{n_a}$ and $\boldsymbol{x}_b \in \mathbb{R}^{n_b}$:

$$\begin{bmatrix} \boldsymbol{x}_a \\ \boldsymbol{x}_b \end{bmatrix} \sim MVT_n\left(\nu, \begin{bmatrix} \boldsymbol{\mu}_a \\ \boldsymbol{\mu}_b \end{bmatrix}, \begin{bmatrix} \boldsymbol{K}_{aa} & \boldsymbol{K}_{ab} \\ \boldsymbol{K}_{ba} & \boldsymbol{K}_{bb} \end{bmatrix}\right),$$

the conditional distribution $p(\boldsymbol{x}_b | \boldsymbol{x}_a)$ is given by:

$$p(\boldsymbol{x}_b | \boldsymbol{x}_a) = MVT_{n_b}(\nu + n_a, \tilde{\boldsymbol{\mu}}_{\boldsymbol{b}}, \frac{\nu + \beta_a - 2}{\nu + n_a - 2}\tilde{\boldsymbol{K}}_{bb}), \tag{2}$$

where

$$\tilde{\boldsymbol{\mu}}_{\boldsymbol{b}} = \boldsymbol{K}_{ba}\boldsymbol{K}_{aa}^{-1}(\boldsymbol{x}_a - \boldsymbol{\mu}_a) + \boldsymbol{\mu}_b$$
$$\beta_a = (\boldsymbol{x}_a - \boldsymbol{\mu}_a)^T\boldsymbol{K}_{aa}^{-1}(\boldsymbol{x}_a - \boldsymbol{\mu}_a)$$
$$\tilde{\boldsymbol{K}}_{bb} = \boldsymbol{K}_{bb} - \boldsymbol{K}_{ba}\boldsymbol{K}_{aa}^{-1}\boldsymbol{K}_{ab}.$$

Derivation of this result is given in the appendix of [8]. Let us now simplify these equations for the case of exchangeable sequences with $\boldsymbol{\mu} = (\mu, \mu \ldots \mu)$ and the following covariance structure:

$$
\boldsymbol{K} = \begin{pmatrix} v & \rho & \cdots & \rho \\ \rho & v & \cdots & \rho \\ \vdots & \vdots & \ddots & \vdots \\ \rho & \rho & \cdots & v \end{pmatrix}.
$$

In our problem, we are interested in doing one-step predictions, i.e. computing a univariate density $p(x_{n+1}|x_{1:n})$ with parameters $\nu_{n+1}$, $\mu_{n+1}$, $v_{n+1}$. Therefore, in Eq. 2 we can take: $n_b = 1$, $n_a = n$, $\boldsymbol{x}_a = x_{1:n} \in \mathbb{R}^n$, $\boldsymbol{x}_b = x_{n+1} \in \mathbb{R}$, $\boldsymbol{K}_{aa} = \boldsymbol{K}_{1:n,1:n}$, $\boldsymbol{K}_{ab} = \boldsymbol{K}_{1:n,n+1}$, $\boldsymbol{K}_{ba} = \boldsymbol{K}_{n+1,1:n}$ and $\boldsymbol{K}_{bb} = \boldsymbol{K}_{n+1,n+1} = v$.

Computing the parameters of the predictive distribution requires the inverse of $\boldsymbol{K}_{aa}$, which we can find using the Sherman-Morrison formula:

$$
\boldsymbol{K}_{aa}^{-1} = (\boldsymbol{A} + \boldsymbol{u}\boldsymbol{v}^T)^{-1} = \boldsymbol{A}^{-1} - \frac{\boldsymbol{A}^{-1}\boldsymbol{u}\boldsymbol{v}^T\boldsymbol{A}^{-1}}{1 + \boldsymbol{v}^T\boldsymbol{A}^{-1}\boldsymbol{u}},
$$

with

$$
\boldsymbol{A} = \begin{pmatrix} v - \rho & 0 & \cdots & 0 \\ 0 & v - \rho & \cdots & 0 \\ \vdots & \vdots & \ddots & \vdots \\ 0 & 0 & \cdots & v - \rho \end{pmatrix},
$$

$$
\boldsymbol{u} = \begin{pmatrix} \rho \\ \rho \\ \vdots \\ \rho \end{pmatrix}, \quad \boldsymbol{v} = \begin{pmatrix} 1 \\ 1 \\ \vdots \\ 1 \end{pmatrix}.
$$

After a few steps, the inverse of $\boldsymbol{K}_{aa}$ is:

$$
\boldsymbol{K}_{aa}^{-1} = \begin{pmatrix} a_n & b_n & \cdots & b_n \\ b_n & a_n & \cdots & b_n \\ \vdots & \vdots & \ddots & \vdots \\ b_n & b_n & \cdots & a_n \end{pmatrix}
$$

with

$$
a_n = \frac{v + \rho(n - 2)}{(v - \rho)(v + \rho(n - 1))},
$$

$$
b_n = \frac{-\rho}{(v - \rho)(v + \rho(n - 1))}.
$$

Note that entries of $\boldsymbol{K}_{aa}^{-1}$ explicitly depend on $n$.

Equations for the mean and variance of the predictive distribution require the following term:

$$
\boldsymbol{K}_{ba}\boldsymbol{K}_{aa}^{-1} = (\rho \quad \rho \quad \cdots \quad \rho)\,\boldsymbol{K}_{aa}^{-1} = \left\{ \frac{\rho}{v + \rho(n - 1)} \right\}_{1:n},
$$

which is a $1 \times n$ vector.

With this in mind, it is easy to derive the following recurrence:

$$
d_n = \frac{\rho}{v + \rho(n - 1)}
$$

$$
\mu_{n+1} = (1 - d_n)\mu_n + d_n x_n
$$

$$
v_{n+1} = (1 - d_n)v_n + d_n(v - \rho).
$$

Finally, let us derive recurrent equations for $\beta_{n+1} = (\boldsymbol{x}_a - \boldsymbol{\mu}_a)^T \boldsymbol{K}_{aa}^{-1} (\boldsymbol{x}_a - \boldsymbol{\mu}_a)$.

Let $\tilde{\boldsymbol{x}} = \boldsymbol{x}_a - \boldsymbol{\mu}_a$, then:

$$\beta_{n+1} = \tilde{\boldsymbol{x}}^T K_{aa}^{-1} \tilde{\boldsymbol{x}}$$

$$= (a_n \tilde{x}_1 + b_n \sum_{i \neq 1}^{n} \tilde{x}_i, a_n \tilde{x}_2 + b_n \sum_{i \neq 2}^{n} \tilde{x}_i, \ldots, a_n \tilde{x}_n + b_n \sum_{i \neq n}^{n} \tilde{x}_i)^T (\tilde{x}_1, \tilde{x}_2, \ldots \tilde{x}_n)$$

$$= (a_n - b_n) \sum_{i=1}^{n} \tilde{x}_i^2 + b_n (\sum_{i=1}^{n} \tilde{x}_i)^2.$$

Similarly, $\beta_n$ from $p(x_n | x_{1:n-1})$ is:

$$\beta_n = (a_{n-1} - b_{n-1}) \sum_{i=1}^{n-1} \tilde{x}_i^2 + b_{n-1} (\sum_{i=1}^{n-1} \tilde{x}_i)^2$$

$$\beta_{n+1} = (a_n - b_n)(\sum_{i=1}^{n-1} \tilde{x}_i^2 + \tilde{x}_n^2) + b_n (\sum_{i=1}^{n} \tilde{x}_i)^2$$

$$= (a_n - b_n) \frac{\beta_n - b_{n-1}(\sum_{i=1}^{n-1} \tilde{x}_i)^2}{a_{n-1} - b_{n-1}} + (a_n - b_n)\tilde{x}_n^2 + b_n (\sum_{i=1}^{n} \tilde{x}_i)^2.$$

It is easy to show that $\frac{a_n - b_n}{a_{n-1} - b_{n-1}} = 1$, so $\beta_{n+1}$ can be written recursively as:

$$s_{n+1} = s_n + \tilde{x}_n$$

$$\beta_{n+1} = \beta_n + (a_n - b_n)\tilde{x}_n^2 + b_n(s_{n+1}^2 - s_n^2),$$

with $s_1 = 0$.

## C   Implementation details

For simple datasets, such as MNIST, we found it tolerable to use models that rely upon a general implementation of the Real NVP coupling layer similarly to Papamakarios et al. [6]. Namely, when scaling and translation functions $s$ and $t$ are fully-connected neural networks. In our model, networks $s$ and $t$ share the parameters in the first two dense layers with 1024 hidden units and ELU nonlinearity [1]. Their output layers are different: $s$ ends with a dense layer with $\tanh$ and $t$ ends with a dense layer without a nonlinearity. We stacked 6 coupling layers with alternating the indices of the transformed dimensions between odd and even as described by Dinh et al. [2]. For the first layer, which implements a logit transformation of the inputs, namely $f(x) = \text{logit}(\alpha + (1 - 2\alpha)x)$, we used $\alpha = 10^{-6}$. The logit transformation ensures that when taking the inverse mapping during sample generation, the outputs always lie within $(\frac{-\alpha}{1-2\alpha}, \frac{1-\alpha}{1-2\alpha})$.

In Omniglot, Fashion MNIST and CIFAR-10 experiments, we built upon a Real NVP model originally designed for CIFAR-10 by Dinh et al. [3]: a multi-scale architecture with deep convolutional residual networks in the coupling layers. Our main difference was the use of coupling layers with fully-connected $s$ and $t$ networks (as described above) placed on top of the original convolutional Real NVP model. We found that adding these layers allowed for a faster convergence and improved results. This is likely due to a better mixing of the information before the output of the Real NVP gets into the Student-t layer. We also found that using weight normalisation [7] within every $s$ and $t$ function was crucial for successful training of large models.

The model parameters were optimized using RMSProp [9] with a decaying learning rate starting from $10^{-3}$. Trainable parameters of a $\mathcal{TP}$ or $\mathcal{GP}$ were updated with a 10x smaller learning rate and were initialized as following: $\nu^d = 1000$, $v^d = 1.$, $\rho^d = 0.1$ for every dimension $d$. The mean $\mu^d$ was fixed at 0. For the Omniglot model, we used a batch size of 32, sequence length of 20 and trained for 200K iterations. The other models were trained for a smaller number of iterations, i.e. ranging from 50K to 100K updates.

# D  Sampling from a Student-t distribution

---

**Algorithm 1** Efficient sampling on GPU from a univariate t-distribution with mean $\mu$, variance $v$ and degrees of freedom $\nu$

---

**function** sample($\mu, v, \nu$)
    $a, b \leftarrow \mathcal{U}(0, 1)$
    $c \leftarrow \min(a, b)$
    $r \leftarrow \max(a, b)$
    $\alpha \leftarrow \frac{2\pi c}{r}$
    $t \leftarrow \cos(\alpha)\sqrt{(\nu/r^2)(r^{-4/\nu} - 1)}$
    $\sigma \leftarrow \sqrt{v\left(\frac{\nu-2}{\nu}\right)}$
    **return** $\mu + \sigma t$
**end function**

---

# E  Sample analysis

In Figure 1, which includes Figure 2 from the main text, we want to illustrate how sample variability depends on the variance of the inputs. From these examples, we see that in the case of a repeated input image, samples get more coherent as the number of conditioning inputs grows. It also shows that BRUNO does not merely generate samples according to the inferred class label.

While Omngilot is limited to 20 images per class, we can experiment with longer sequences using CIFAR-10 or MNIST. In Figure 2 and Figure 3, we show samples from the models trained on those datasets. In Figure 4, we also show more samples from the prior distribution $p(\boldsymbol{x})$.

Figure 1: Samples generated conditionally on images from an unseen Omniglot character class. *Left:* input sequence of 20 images from one class. *Right:* the same image is used as an input at every step.

Figure 2: CIFAR-10 samples from $p(\mathbf{x}|\mathbf{x}_{1:n})$ for every $n = 480, \ldots, 500$. *Left*: input sequence (given in the top row of each subplot) is composed of random same-class test images. *Right*: same image is given as input at every step. In both cases, input images come from the test set of CIFAR-10 and the model was trained on all of the classes.

Figure 3: MNIST samples from $p(\mathbf{x}|\mathbf{x}_{1:n})$ for every $n = 480, \ldots, 500$. *Left*: input sequence (given in the top row of each subplot) is composed of random same-class test images. *Right*: same image is given as input at every step. In both cases, input images come from the test set of MNIST and the model was trained only on even digits, so it did not see digit '1' during training.

Figure 4: Samples from the prior for the models trained on Omniglot, CIFAR-10, Fashion MNIST and MNIST (only trained on even digits).

# F  Parameter analysis

After training a model, we observed that a majority of the processes in the latent space have low correlations $\rho^d/v^d$, and thus their predictive distributions remain close to the prior. Figure 5 plots the number of dimensions where correlations exceed a certain value on the x-axis. For instance, MNIST model has 8 dimensions where the correlation is higher than 0.1. While we have not verified it experimentally, it is reasonable to expect those dimensions to capture information about visual features of the digits.

Figure 5: Number of dimensions where $\rho^d/v^d > \epsilon$ plotted on a double logarithmic scale. *Left*: Omniglot model. *Middle*: CIFAR-10 model *Right*: Non-convolutional version of BRUNO trained on MNIST.

For $\mathcal{TP}$-based models, degrees of freedom $\nu^d$ for every process in the latent space were intialized to 1000, which makes a $\mathcal{TP}$ close to a $\mathcal{GP}$. After training, most of the dimensions retain fairly high degrees of freedom, but some can have small $\nu$'s. One can notice from Figure 6 that dimensions with high correlation tend to have smaller degrees of freedom.

Figure 6: Correlation $\rho^d/v^d$ versus degrees of freedom $\nu^d$ for every $d$. Degrees of freedom on the x-axis are plotted on a logarithmic scale. *Left*: Omniglot model. *Middle*: CIFAR-10 model *Right*: Non-convolutional version of BRUNO trained on MNIST.

We noticed that exchangeable $\mathcal{TP}$s and $\mathcal{GP}$s can behave differently for certain settings of hyperparameters even when $\mathcal{TP}$s have high degrees of freedom. Figure 7 gives one example when this is the case.

Figure 7: A toy example which illustrates how degrees of freedom $\nu$ affect the behaviour of a $\mathcal{TP}$ compared to a $\mathcal{GP}$. Here, we generate one sequence of 100 observations from an exchangeable multivariate normal disribution with parameters $\mu = 0.$, $v = 0.1$, $\rho = 0.05$ and evaluate predictive probabilities under an exchangeable $\mathcal{TP}$ and $\mathcal{GP}$ models with parameters $\mu = 0.$, $v = 1.$, $\rho = 0.01$ and different $\nu$ for $\mathcal{TP}$s in the left and the right plots.

# G  Training of $\mathcal{GP}$ and $\mathcal{TP}$-based models

When jointly optimizing Real NVP with a $\mathcal{TP}$ or a $\mathcal{GP}$ on top, we found that these two versions of BRUNO occasionally behave differently during training. Namely, with $\mathcal{GP}$s the convergence was harder to achive. We could pinpoint a few determining factors: **(a)** the use of weightnorm [7] in the Real NVP layers, **(b)** an intialisation of the covariance parameters, and **(c)** presence of outliers in the training data. In Figure 8, we give examples of learning curves when BRUNO with $\mathcal{GP}$s tends not to work well. Here, we use a convolutional architecture and train on Fashion MNIST. To simulate outliers, every 100 iterations we feed a training batch where the last image of every sequence in the batch is completely white.

We would like to note that there are many settings where both versions of BRUNO diverge or they both work well, and that the results of this partial ablation study are not sufficient to draws general conclusions. However, we can speculate that when extending BRUNO to new problems, it is reasonable to start from a $\mathcal{GP}$-based model with weightnorm, small initial covariances, and small learning rates. However, when finding a good set of hyperparameters is difficult, it might be worth trying the $\mathcal{TP}$-based BRUNO.

Figure 8: Negative log-likelihood of $\mathcal{TP}$ and $\mathcal{GP}$-based BRUNO on the training batches, smoothed using a moving average over 10 points. *Left:* not using weightnorm, initial covariances are sampled from $\mathcal{U}(0.1, 0.95)$ for every dimension. Here, the $\mathcal{GP}$-based model diverged after a few hundred iterations. Adding weighnorm fixes this problem. *Middle:* using weightnorm, covariances are initialised to 0.1, learning rate is 0.002 (two times the default one). In this case, the learning rate is too high for both models, but the $\mathcal{GP}$-based model suffers from it more. *Right:* using weightnorm, covariances are initialised to 0.95.

# H  Set anomaly detection

Online anomaly detection for exchangeable data is one of the application where we can use BRUNO. This problem is closely related to the task of content-based image retrieval, where we need to rank an image $\boldsymbol{x}$ on how well it fits with the sequence $\boldsymbol{x}_{1:n}$ [5]. For the ranking, we use the probabilistic score proposed in Bayesian sets [4]:

$$\text{score}(\boldsymbol{x}) = \frac{p(\boldsymbol{x}|\boldsymbol{x}_{1:n})}{p(\boldsymbol{x})}. \tag{3}$$

When we care exclusively about comparing ratios of conditional densities of $\boldsymbol{x}_{n+1}$ under different sequences $\boldsymbol{x}_{1:n}$, we can compare densities in the latent space $\mathcal{Z}$ instead. This is because the Jacobian from the change of variable formula does not depend on the sequence we condition on.

For the following experiment, we trained a small convolutional version of BRUNO only on even MNIST digits (30,508 training images). In Figure 9, we give typical examples of how the score evolves as the model gets more data points and how it behaves in the presence of inputs that do not conform with the majority of the sequence. This preliminary experiment shows that our model can detect anomalies in a stream of incoming data.

Figure 9: Evolution of the score as the model sees more images from an input sequence. Identified outliers are marked with vertical lines and plotted on the right in the order from top to bottom. Note that the model was trained only on images of even digits. *Left:* a sequence of digit '1' images with one image of '7' correctly identified as an outlier. *Right:* a sequence of digit '9' with one image of digit '5'.

# I Model samples

Figure 10: Samples from a model trained on Omniglot. Conditioning images come from character classes that were not used during training, so when $n$ is small, the problem is equivalent to a few-shot generation.

Figure 11: Samples from a model trained on CIFAR-10. The model was trained on the set with 10 classes. Conditioning images in the top row of each subplot come from the test set.

Figure 12: Samples from a convolutional BRUNO model trained on Fashion MNIST. The model was trained on the set with 10 classes. Conditioning images in the top row of each subplot come from the test set.

Figure 13: Samples from a non-convolutional model trained on MNIST. The model was trained on the set with 10 classes. Conditioning images in the top row of each subplot come from the test set.