[Reviews · NeurIPS 2018]

Reviewer 1



This paper introduces an unsupervised approach to modeling exchangeable data. The proposed method learns an invertible mapping from a latent representation, distributed as correlated-but-exchangeable multivariate-t RVs, to an implicit data distribution that can be efficiently evaluated via recurrent neural networks. I found the paper interesting and well-written. Justification and evaluation of the method could, however, be much better. In particular, the authors do not provide good motivation for their choice of a multivariate t-distribution beyond the standard properties that 1) the posterior variance is data-dependent 2) it is heavier tailed compared to normal. The experiment in Sec. 4.3 provides some evidence that this choice is effective, but a qualitative comparison on a single instance is insufficient to draw any real conclusions. I am not sure I follow the construction in Sec. 4.2. Is a conditional model learned within each class? If not then I don't follow the model comparison approach to classification. If this is true, then I don't know that much of the discussion about the limitation of generative models for this task (L:253-262) is really relevant or at least should be revised to reflect the reality that it isn't a fully generative model. Please clarify. The goal for the conditional image generation experiment in Sec. 4.1. seems to be a demonstration that samples from the posterior predictive distribution of BRUNO have higher variability when conditioned on more variable inputs. While this seems like a reasonable property I am not convinced that an experiment dedicated to this is an effective use of space. Moreover, the authors only present a single qualitative result in Fig. 2., from which it isn't clear that variance of Fig. 2. right is actually lower. Why not report quantitative results by computing the empirical covariance over (more) samples from the model? Also, if the authors could not reproduce results from [5] it would be entirely appropriate to state this and report comparison with the neural statistician using results they do have. Some detailed comments below: * Aliasing artifacts make Fig. 2 right appear as if there are a couple different input images. Consider enlarging image or using a vector graphic. * What are the learned degrees of freedom \nu in Sec. 4.3? Are they small so that the prior p(z) is far from Gaussian? How do these compare to learned DoFs for other experiments? * It isn't directly shown that the assumed MVT covariance leads to exchangeability. I assume this is straightforward from the Gaussian example and the continuous mixture representation of MVT?

Reviewer 2



[After author rebuttal] My thanks to the authors for their careful rebuttal. I found it convincing & I've modified my review to a weak accept. Original review below. [Original review] This paper proposes a model for exchangeable data based on sampling latent variables from an exchangeable student T process, followed by passing the variables through an easily-invertible (NVP) transformation. The paper proposes an efficiently-computable decomposition of the proposed distribution, and provides experimental evidence of its utility. This paper is reasonably well-written and clearly explained. The intro and relevant work sections are particularly well done. Some comments for improvements in the rest of the paper: -- section 3.3: the generative model is described backwards, i.e. x first (with x exchangeable), then a mapping from z to x (z exch), then the distribution on z. I would think it would be easier to understand if the distribution on z is described, then how z maps to observed data x, then show x is exchangeable -- from l.173-183, it's clear that the paper derives the quantities needed to express p(z_{n+1} | z_{1:n}), but it's not clear why that distribution is needed or useful -- section 3.4 makes it clearer that the efficient expression of p( *_n+1 | *_{1:n} ) is used for training, where the goal is to maximize the log marginal p(x_1, ... x_n) (decomposed as the sequence of conditionals). -- I think the right way to fix this confusion is to move section 3.4 earlier in 3.3, then expand on it. That is, state around l.168 that given the specification of the model, the major task is to train it. Then explicitly write the training problem of maximizing the log marginal, then decompose it, then show that the student T process yields a nice efficient formula for the decomposition, then argue this can be optimized (via e.g. autograd). Finally, mention that the efficient conditional formula can be used for prediction once training is done. It would also be very helpful to write down the full model including the NVP transformation and how it affects training. -- A similar clarity issue occurs in the experiments. It's generally unclear how data is being used for training & testing/prediction. Is the idea that you take a bunch of sequences of length N and optimize the sum of log marginals for each ( so maximize sum_{j=1}^J sum_{n=1}^{N-1} log p(x_{j,n+1} | x_{j,1:n}) where each (x_{j,1}, ... , x{j, N}) is an observed sequence )? -- I'm not sure what I'm supposed to get from figure 2. Aside from being too small (given the intricate characters), it is missing a lot of explanation. What is an "input sequence"? Were these samples generated on a model trained on the whole dataset or a single sequence? Is this showing samples of p(x_{N+1}, ... x_{2N} | x_{1:N}) where x_{1:N} are shown in the top row? Is this all simulated from the model? I really am not sure what I'm seeing here. -- I'm also not convinced that I see "lower variability in appearance" for images in the right col -- the experiment in 4.2 is a bit clearer, but it's still missing details on how the model was trained. Is this the same model from 4.1? -- the extra "discriminative training" is not clear and not well-described. - a technical question: Is there a known de Finetti representation for the exchangeable student T (similar to that for the gaussian at l.37)? If one were to explicitly incorporate this top hierarchical layer into the model, would that make inference more efficient (given that the z's would then be conditionally independent)? Given the missing/unclear details in section 3 and the experiments, I've given this a weak reject. It's very borderline though; I'd certainly be open to switching to an accept given a convincing rebuttal. Detailed comments: - l.2 - the paper asserts it performs exact bayesian inference. I would hesitate to make this claim, given that the paper marginalizes out the latent parameter - l.12-16: exchangeable sequences are not a model for unordered sets, but rather for ordered sequences where the probability of generating a particular sequence is invariant to reordering. Sequences of unordered sets (and derivative combinatorial structures) need not be exchangeable, although sometimes are. cf Pitman, "exchangeable and partially exchangeable random partitions" and Broderick, Pitman, & Jordan "feature allocations, probability functions, and paintboxes" - Figure 1 should appear later in the paper, perhaps at the beginning of section 3.3. - mathcal script font shouldn't be used for TP / GP. Just use regular text for these initialisms - l.105: "nontrivial differences in the task" - better to just be clear here about exactly why they're different and what benefit TP gives over GP - l.130: In particular it's the marginalization over the inverse Wishart covariance matrix - l.134: NVP is undefined - l.150-153: is the uniform noise discussion really necessary? Why not just ignore the discreteness and treat the quantized data as real-valued in practice? - l.163+: generally I'd just use two subscripts rather than superscript - too easily confused with exponentiation (especially in statements like K_{ii}^d = v^d)

Reviewer 3



After rebuttal: I am happy with the rebuttal and the authors have addressed all my concerns. Therefore, I will still recommend accepting this paper. Before rebuttal: Working with exchangeable data, the authors propose a Bayesian method called BRUNO to construct posterior predictive distribution. The main idea is to first map observations unto the latent space by stacking coupling layers consisting of neural networks, where this map is bijective and it preserves the exchangeability property. Once in the latent space, they assume that the transformed data follows a student-t process, and they derived efficient recurrent equations to construct the resulting latent posterior predictive distribution. When predictions are desired, exchangeable samples are drawn from this latent predictive distribution, and they are coupled with the aforementioned map to obtain the corresponding predictions in the original space. This paper is very well written and the main ideas are presented clearly. The authors show that their method can be applied in many application domains, such as conditional image generation, few-shot learning and online anomaly detection; and they conducted extensive numerical experiments to further show that it is competitive with respect to the state-of-the-art, at least for the few-shot learning task example. The success of the method appears to depend crucially on using student-t distribution to make predictions, and they gave detailed explanation as to why this is superior to the more commonly used Gaussian processes. On top of that, exchangeability in the latent space allows one to derive updating equations with computational complexity that is linear in the dimension, and this allows one to bypass doing matrix inversion if the usual full hierarchical Bayesian approach is used. Consequently, I find that this work is a very nice contribution to the literature. My first comment concerns the interpretation of the generated images (e.g., Figure 2 in the main text and Figures 3-6 in Appendix Section G). For the MNIST data (Figure 6), it is clear that BRUNO is producing good results as the conditionally generated images are the same as the conditioned input images, albeit with some slight (stochastic) variations. However I notice that in the CIFAR10 example (Figure 4), there are a lot of green patches and I am curious to know what are they? Also in this example, what constitutes a reasonable performance? For example, if the conditioned images are horses, then the generated images must have horses in them with some variations? I am having a hard time to recognize the generated images for CIFAR10 given the input images. In fact how does BRUNO compare with other generative algorithms in generating images such as Generative Adversarial Network (GAN)? As exchangeability seems to feature prominently in this paper, a natural question then is whether BRUNO will work for non-exchangeable data, or whether it can be extended to this case? Here are some typos and further comments: 1. Line 35 "which of the random variables" 2. Line 98, because NVP is mentioned many times in this paper, I think you should at least write what NVP represents in this line. 3. Lines 122-123, $\beta$ should be $\beta_a$ 4. Line 148, $Z$ should be $\mathcal{Z}$ 5. Line 174, explain that $\beta_1$ here is the initialization of $\beta_a$ in Eq. 6 6. In both the main paper and the appendices, the citation format is not consistent, some are numbers but others are names and numbers. 7. In both the main paper and the appendices, correct punctuation (commas or periods) should be placed at the end of displayed equations, e.g., see the display in Line 180 and many others. 8. In Figure 3, the label "theor" is theoretical (i.e., the assumed prior)? Maybe it is better to clarify this in the caption. 9. For Appendix A, it seems that Lemma 1 is not used in the proof of Lemma 2, or anywhere in the paper as far I as know. So I think Lemma 1 can be dropped from the paper. 10. In Line 24 of Appendix B, I would write "simpler" instead of "easier". 11. Line 35 Appendix B, I think it is better to give a reference to the Sherman-Morrison formula, so that people who are not familiar can look it up. 12.In Line 50 of Appendix B, the $\beta_{n+1}$ equation does not seem to be correct, see the display in Line 180 in the main text. Also, $s_1=0$. 13. Line 59 of Appendix C, no need "the the" 14. Line 61, what is $\alpha$ here? 15. Line 105, I think it is better to clarify the right plot of Fig. 1 in Section E. Also it appears that the left plot of Fig. 1 is not discussed in the few-shot learning experiments in the main text. 16. At at my first read, I did not understand the schematics in Figure 1 and many expressions and work flows will only be explained in subsequent sections. However, subsequent sections do not refer back to Figure 1. Therefore, I suggest that this Figure should be moved to a place after BRUNO has been explained, or at least make frequent references to it when discussing BRUNO.